# Large Scale Optical Assisted Mm-Wave Beam-Hopping System for Multi-Hop Bent-Pipe LEO Satellite Networks

**Shiyi Xia** [1,2,3] , **Peilong Liu** [4] , **Mingyang Zhao** [5] , **Cheng Zou** [1,2] , **Fengwei Shao** [1,2] , **Jifeng Jin** [1,2] , **Haiwang Wang** [1,2] and **Guotong Li** [1,2,*]

1    Innovation Academy for Microsatellites of CAS, Shanghai 201203, China
2    University of Chinese Academy of Sciences, Beijing 100049, China
3    Institute of Photonic Integration, Eindhoven University of Technology,
     NL 5600 MB Eindhoven, The Netherlands
4    Beijing National Research Center for Information Science and Technology, Tsinghua University,
     Beijing 100190, China
5    International Collaborative Laboratory of 2D Materials for Optoelectronics Science Technology of Ministry of
     Education, Institute of Microscale Optoelectronics, Shenzhen University, Shenzhen 518060, China
*    Correspondence: ligt@microsate.com

**Abstract:** By introducing the arrayed waveguide router (AWGR) optical true time-delay (OTTD) architecture in bent-pipe satellite optical inter-link to optically assist the RF phased array in Low-earth orbit satellites will extend the multi-hop bent-pipe satellite beam-hopping protocol proposed in DVB-S2X. It solves the challenge of beam steering with the support of precise, broadband, and wide-range scanning. This architecture utilizes a subarray to combine the advantages of AWGR and a high-precision RF phase shifter to realize the beam pointing without an oblique view. Unlike the traditional digital and analog phased array architecture, the introduction of OTTD can solve the problem of beam squint and also ensure the high-precision scanning of the beam.

**Keywords:** LEO satellite constellation; bent-pipe satellite; beamforming; beam-hopping; optical true time dealy

## 1. Introduction

Satellites are attractive to both developed and developing countries because of their wide coverage. Low-earth orbit(LEO) satellite networks enable ubiquitous wireless coverage, which facilitates information access in rural areas of terrestrial networks. LEO satellites serve a wide range of audiences, including aircraft, ships, and automobiles, which in turn, placed a high demand on beam flexibility. According to the next-generation communication protocol [1–3], satellite internet is defined as a complement to the terrestrial internet. It benefits from LEO satellites' low orbital height, inexpensive launch, speedy transmission, and low path loss [4].

In the post-5G era, satellites are required to provide broadband services under large-scale constellations without interfering with other links, which requires flexible and controllable satellite beams in terms of directivity [5,6]. Due to their outstanding dependability, low cost, and simplicity in implementing numerous beams, phased arrays have replaced classic mechanical scanning antennas [7,8]. A number of phased-array antennas have been made available for satellite use in terms of beam forming, beam switching, and large beam-steering angle [9–13]. This high flexibility also makes the beam-hopping proposed in the DVB-S2X protocol possible [14].

For instance, each of the three active phased-array antennas on the US Iridium/NEXT constellation's LEO satellites can produce 16 beams when operating in the L-band [15]. The current major large LEO satellite constellation SpaceX also uses phased-array antennas as well as OneWeb and Telesat. Each SpaceX satellite is outfitted with a sophisticated digital

payload that contains a phased array, allowing for the independent steering and shaping of each beam. Each of the 16 very elliptical beams that OneWeb produces, which correspond to the satellite's flight path, leaves an imprint on Earth. With individually formable and programmable beams that may be sparsely deployed and precoded to serve customers, Telesat and SpaceX are comparable [16,17].

It is generally known that huge LEO satellites' constellations can interact directly via laser links, making the compatibility of phased array systems for LEO satellites with optical links all the more crucial [18]. First, in 1995 Frigyes et al. used optical techniques to solve various approaches to the fundamental problem of broadband phased arrays, namely the most critical beam strabismus [19]. Lee et al. proposed an optical true-time delayed (OTTD) beamforming system using dispersion-compensating fiber and multi-wavelength lasers for phased-array antennas in 2011 [20]. Starlink is an existing electronically controlled phased-array antenna system that is extremely scalable and aircraft independent. They designed and manufactured a new all-optical driven passive phase shifter in the X-band, based on a nematic phase liquid crystal mesocrystal azo dye hybrid in 2020 [21].

Due to the recent rapid advancement of photonic integration technology, some integrated OTTD has undergone several modifications and iterations with noticeable performance increases [22–24], especially optical true delay lines based on the array waveguide grating router(AWGR) structure [25–30]. Dr Cao built and characterized an array waveguide grating feedback loop-based remotely controllable integrated OTTD (AWG loop). Since AWGs are employed as wavelength multiplexers (MUXs) and de-multiplexers (de-MUXs), AWG loops can offer an integrated OTDL solution that is small, fabrication-resistant, and scalable [31]. Zhang et al. reported an AWGR integrated chip with doubled delay resolution and experimentally verified its use in a 38-GHz fiber-optic wireless beam-steering system with 186° angular steering [32].

In a multi-hop bent-pipe LEO satellite network, a flexible beam-management system especially created to include an optical switching network is essential. The technology requires extremely precise beam directing and is compatible with low-complexity, easy-to-manage satellite multi-wavelength optical switching designs. As shown in Figure 1, we propose a novel optical controlled phased array for non-regenerative LEO satellite networks. To our knowledge, it is the first architecture proposed to support multi-hop bent-pipe satellite networks using the DVB-S2X beam-hopping protocol [14]. Thanks to AWGR-assisted millimeter wave-phased array, we make it possible for transparent satellites to achieve beam hopping. We investigate in detail an AWGR-controlled phased array construction that satisfies the requirements of the constellation. The AWGR-based optically assisted phased array may provide a number of cutting-edge features for broadband millimeter-wave communications: in order to extend interplanetary routing over the air to ground coverage, satellite network designs can easily incorporate wavelength division multiplexing, which allows for a skew-free beam scan at large angular beam pointing, ultra-high precision beam scan intervals, adjustable beam pointing, and reduced satellite processing complexity redundancy.

We provide insight into LEO satellite network requirements in Section 2. In Section 3, the idea of using an optically controlled phased array-based subarray was proposed and we theoretically demonstrate its enhancement of the bandwidth performance of broadband phased arrays. The feasibility of the optically controlled millimeter wave phased array structure based on the arrayed waveguide grating routers (AWGR) structure is experimentally demonstrated in Section 4. Section 5 provides a summary and outlook.

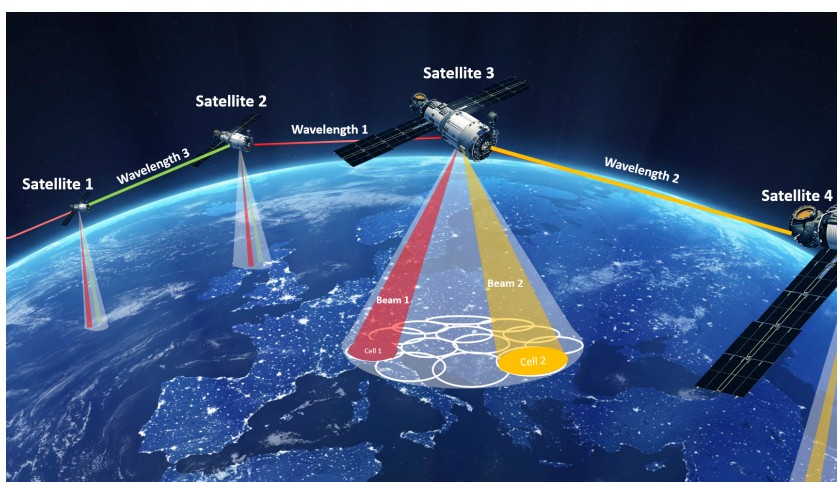

**Figure 1.** Low-orbiting satellite network employs inter-satellite laser communication and optically controlled millimeter wave-phased array. Wavelength multiplexing between constellations is used to transfer data from users in different cells. The wavelength multiplexing can be directly converted into the corresponding millimeter wave-beam pointing by unique characteristics of the array waveguide grating router structure.

## 2. Multi-Hop Bent-Pipe LEO Satellite Constellation

Benefiting from optical ISL, low latency, and low-noise optical switching via wavelength multiplexing(WDM) technology, optically transparent transmission channels can be established between satellites, which can be used to eliminate the expensive regeneration and packet processing resources. Furthermore, the satellite internet system can use the large aperture antenna of the gateway station to compensate for the deterioration of the SNR.

To improve the flexibility of optical satellite networks, the proven WDM technology on the optical networks can be introduced very effectively to create multiple optical path channels in the ISL [33]. As shown in Figure 2, after a flow is transmitted toward the satellite through the uplink channel, it will be modulated into the optical signal of a specific wavelength, and forwarded by the optical switch according to the two-tuple entry (ingress port, wavelength) recorded in the onboard forwarding table. Finally, the flow can be forwarded to the destination in a multi-hop manner.

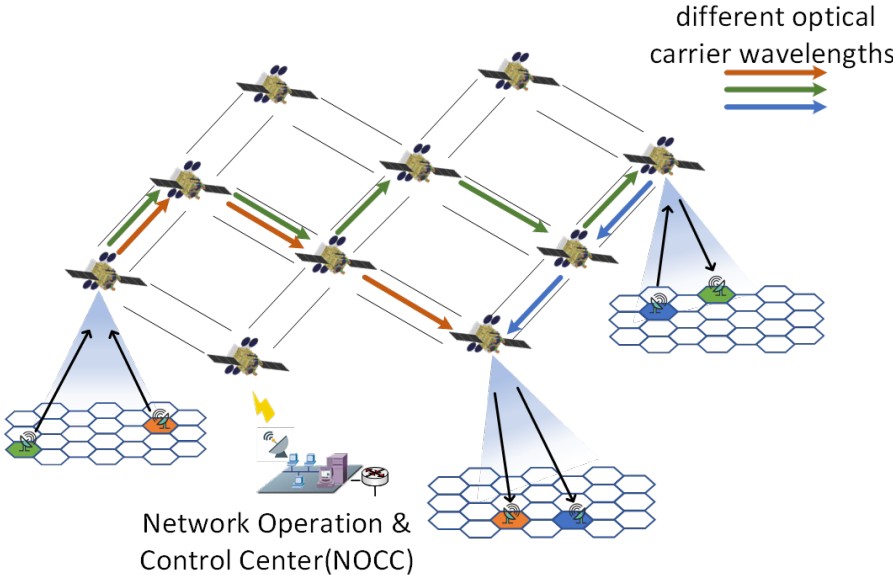

**Figure 2.** Wavelength Assignment and Routing.

### 2.1. Multi-Hop Bent-Pipe Beam-Hopping Payload Architecture

Satellite networks can use WDM ISLs to achieve high resource utilization based on the aforementioned strategy. Therefore, a brand-new design for a satellite payload is shown in Figure 3. Once the optical carrier forwarded from another satellite $\lambda_{ISL}^{in}$ (or sent from the ground $\lambda_{AC}^{in}$ into the payload with wavelengths which carry different RF signals) enters the satellite payload, the wavelength selective switch (WSS) sends the optical carrier signals of the corresponding wavelengths to optically controlled millimeter beamformers $\lambda_{AC}^{out}$ or forwards them to adjacent satellites $\lambda_{ISL}^{out}$ according to the routing table uploaded by the NOCC on the ground. Different beamformers respond to varied wavelengths with different beam-pointing responses, for example, wavelength 1 corresponds to a beam pointing of 10° in beamformer 1 and 20° in beamformer 2. This design expands the single bent-pipe satellite beam hopping technique for DVB-S2X to multiple hops while maximizing the use of LEO satellite resources [14]. The introduction of beam-hopping technology into non-regenerative satellite networks and resolving the issue of high-precision beam pointing of broadband RF signals could add significant benefits to the architecture of LEO satellite networks:

- Saving regenerative processing loads between satellites, making satellites lighter and lower-power;
- Makes it simple to upgrade and alter satellite payload;
- Integrate beam-hopping into satellite network routing algorithms to achieve more flexible inter-satellite routing reconfiguration.

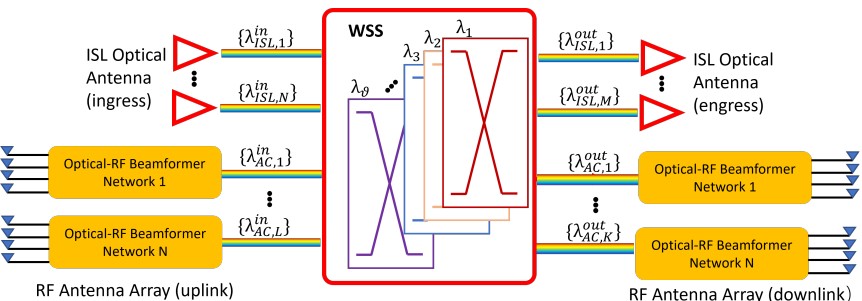

**Figure 3.** Payload architecture.

### 2.2. Antenna Design Challenge for Beam-Hopping

The specific wavelength allocated to each service and the forwarding entry of each wavelength are determined by the wavelength routing algorithm [34–36]. The number of available wavelengths of each laser is limited. Once used up, the ISL can no longer serve new traffic, which may lead to call blocking. Therefore, the wavelength routing algorithms generally focus on minimizing the number of wavelengths used or minimizing the blocking rate.

Flexible, high-gain broadband spot beam access, which allows users to customize their access on demand and is another technical approach for broadband internet satellites, requires precise beam aiming. Over 10 Gbps data transmission service can be achieved through the coverage of the spot beam. This makes it possible to integrate LEO satellite communications with 5G networks.

As shown in Table 1, three representative large LEO satellite networks now in use all employ high-gain point beams [37–39]. The beam widths are all quite near to 1°. In order to achieve full coverage, the pointing interval of the beam must be close to 1°. The beam's control precision must be less than 1° if it is to achieve overlapping coverage. Such high precision control causes the design of the beam control components to face significant challenges. Additionally, broadband signal beamforming techniques must still be used because of the high accuracy standards to prevent the beam squint.

**Table 1.** Three large constellation beam coverage designs.

| Example | No. Beam | Angle | Beam Width | Antenna Type |
|---|---|---|---|---|
| Starlink | 16 | 56.55° | 1.3° | Phase array |
| OneWeb | 16 | 40° | TX:3°; RX:2° | Phase array |
| Telesat | 16 | 45° | 1.55° | Phase array |

There are currently two main methods of achieving accurate beam pointing in broadband communications: digital delay lines and optical true delay lines. The digital delay line achieves delay compensation by sampling the signal with a high-speed ADC. OneWeb is currently working with Satixfy to launch a digital technology path payload [40]. The optical true delay line modulates the RF signal onto the optical carrier, based on the optical delay line or slow light effect, to create a true delay in the RF signal.

## 3. Optical Controlled Phased Sub-Array in Multi-Hop Bent-Pipe Payload

An optically controlled phased array solution for multi-wavelength optical networks is proposed, shown in Figure 4, for multi-hop Bbnt-pipe low-orbit satellite constellations. In order to achieve beam-hopping and precise pointing, the millimeter wave phased array is directly controlled based on the multi-wavelength inter-satellite link optical signal, which simplifies the wave-controlled payloads and incorporates RF payloads into the entire optical switching satellite network.

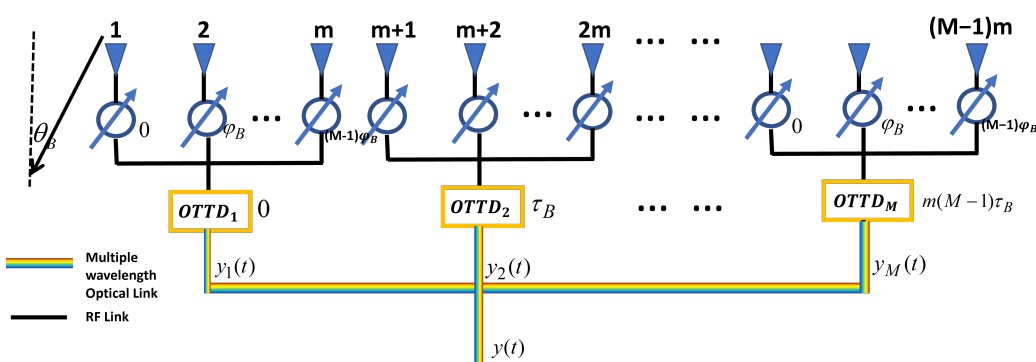

**Figure 4.** Principle of AWGR Assisted Mm-wave Beam-hopping System.

### 3.1. Optically Controlled Beam Steering

The overall phase-shifting network structure is a two-layer phase regulation, with the first layer consisting of $M$ AWGRs, which introduce OTTDs for $M$ subarrays. Each AWGR can generate a different true time delay controlled by the wavelength. The second layer consists of $(M-1)m$ digital phase shifters, who provide the exact phase shift.

The time difference between two adjacent array elements is $\tau = d\sin\theta/c$, where $d$ is the distance between two antenna array elements, $c$ is the light speed, and $\theta$ represents the beam angle. The phase difference is $\varphi = 2\pi d\sin\theta/\lambda_0$, where $\lambda_0$ is the center frequency of the signal. For example, when the beam is pointing at $\theta_B$, the time delay and phase difference can be expressed as $\tau_B = d\sin\theta_B/c$ and $\varphi_B = 2\pi d\sin\theta_B/\lambda_0$, respectively. Based on beamforming theory [41], the far field pattern (FFP) of such an array could be expressed as:

$$AF(\theta) = \sum_{i=0}^{M-1} \sum_{l=0}^{m-1} [exp(-j((im+l)kdsin(\theta))(\alpha_{\lambda_0}(i) + \alpha_B(l)))] \qquad (1)$$

Every subarray element phase difference of the digital phase shifter can be expressed as:

$$\alpha_B(l) = [0, \varphi_B, 2\varphi_B, \dots, (m-1)\varphi_B], l = 0, 1, 2, \dots, m \qquad (2)$$

To be more compatible with satellite communications, the delay design introduced by the AWGR for each sub-array needs to be predefined. For a optical carrier , wavelength $\lambda_0$, the $i^{th}$ subarray true time delay introduced by AWGR is:

$$\alpha_{\lambda_0}(i) = 2\pi c[0, m\tau_B, 2m\tau_B \dots, \tau_B(M-1)m]/\lambda, i = 0, 1, 2, \dots M \tag{3}$$

Such a delay eliminates the delay difference between sub-arrays. In addition, since the delay time of the AWGR is fixed for a specific wavelength, WDM can be converted to space division multiplexing in optical network switching by this structure. And according to our previous research, AWGR allows for delay control of vector synthesis [42]. That is, the delay introduced by $\lambda_0$ can be achieved by simultaneously loading the RF signal into $\lambda_1$ and $\lambda_2$, and the combination of vector synthesis of delay quantities allows for a more refined OTTD.

The scanning angle of the beam is determined by the amount of phase shift of the array. The minimum phase shift of the digital phase shifter is indicated by $2\pi/2^K$. The angle at which the antenna beam is pointed can therefore be expressed as $p$ digital phase shifts and $q$ true time delays

$$\varphi = \frac{2\pi d sin\theta}{\lambda} = p \times 2\pi/2^K + q \times 2\pi\tau_{min}c/\lambda_0 \tag{4}$$

Using a differential solution, the minimum beam accuracy can be obtained

$$\frac{d\varphi}{d\theta} = 2\pi d cos\theta/\lambda \tag{5}$$

$$\Delta\theta = \frac{\Delta\varphi\lambda}{2\pi d cos\theta} = \frac{\lambda}{2^K d cos\theta} \tag{6}$$

The wavelength does not need to be adjusted when fine-tuning the scanning angle, so the phase delay caused by the true time delay generated by AWGR can be ignored when performing high-precision scanning.

Assume the first receiver signal is $s(t) = e^{j\phi(t)}$ The $i^{th}$ subarray synthesis signal is:

$$y_i(t) = s(t) \times exp(-j(\alpha_B(l) + \alpha_{\lambda_0}(i))), i = 1, 2, \dots, m; l = 1, 2, \dots, M \tag{7}$$

The total array synthesis signal is:

$$y(t) = \sum_{i=0}^{M-1} y_i[t - mi\tau_B] \tag{8}$$

Large phase differences are compensated for each sub-array by adjusting the delay amount to $\tau_B$ in $\alpha_s$ by AWGR. Ultra-high precision beam scanning is then achieved by fine-tuning the high-precision digital phase shifter $\alpha_B$ in $\varphi_B$. With such a two-stage phase adjustment, the final array synthesized signal allows both large-angle beam switching and ultra-high accuracy scanning and target tracking.

### 3.2. Broadband Phased-Array Aperture Effect

The broadband phased array aperture effect is mainly characterized by the constraint of the array antenna's crossing time $T_A = Lsin\theta_B$ on the bandwidth of the transmitted signal, expressed as a certain deflection of the beam pointing by the wideband signal. The relationship between the deflection angle and bandwidth of a conventional phased array can be expressed as [43]:

$$\Delta\theta_f = \frac{\Delta f}{f_0}tan\theta_B \tag{9}$$

where $f_0$ is the center frequency of the signal, $\Delta f$ represents the bandwidth of the signal As a result, the beam pointing skew caused by the change in signal frequency increases

as the bandwidth $\Delta f$ and scanning angle become larger. The bandwidth criterion for phased-array antennas is the limit on $\Delta f$. When $\theta_B = 60^o$, usually the maximum beam deflection is called to satisfy

$$\Delta f_{max} \leq \theta_{1/2}(\theta_B)/4. \tag{10}$$

The $\theta_{1/2}(\theta_B)$ is the half power beam width (HPBW) when the beam is scanned to $\theta_B$.

$$\theta_{1/2}(\theta_B) = \theta_{1/2}/cos\theta_B \tag{11}$$

where $\theta_{1/2}$ is the HPBW of the beam in the direction normal to the array plane. Therefore,

$$\frac{\Delta f}{f_0} \leq \theta_{1/2}/4sin\theta_B \tag{12}$$

If the scanning angle $\theta_B = 60^o$ and the HPBW is $2^o$, the limitation on the signal bandwidth is 0.02. For a 30 GHz phased array, the maximum instantaneous signal bandwidth allowed is 600 MHz, which is hardly enough to meet the requirements of current low-orbit communication satellites.

If the antenna is divided equally into $M$ sub-arrays, as shown in the figure, there are m antenna elements within each sub-array. The beam pointing of each subarray is still altered despite the subarray's continued exposure to the transition time and aperture effect; however, because each subarray's aperture represents only $1/M$ of the total array, the subarray's aperture transition time will be decreased to

$$T'_A = T_A/M \tag{13}$$

At this point, the system limit on the instantaneous signal bandwidth can be relaxed by a factor of $M$. Equation (10) will be reduced to

$$\frac{\Delta f}{f_0} \leq M\theta_{1/2}/4sin\theta_B \tag{14}$$

## 4. Experiment and Results

Schematic diagram of the measurement setup as shown in Figure 5. The key components of the experiment are a tunable laser source (TLS photonetics Wavelength Tunable Laser) and two symmetrical AWGRs (Gemfire 80ch AWG DMX). TLS controls the true-time delay value via adjusting the wavelength. Tunable optical carriers generated by the TSL are coupled and fed into a Mach–Zehnder modulator(MZM) via a polarization controller. Since the FSR of the two symmetrical AWGRs is 40 GHz, the maximum modulable RF signal for this structure is 40 GHz. The vector network analyzer (VNA, Keysight FieldFox N9952A) generates millimeter wave RF signals in the frequency range from 25 GHz to 36 GHz. The optical carrier is modulated by the RF signal generated by the VNA . Erbium-doped fiber amplifier (EDFA) is used to compensate for the loss of symmetrical AWGR to ensure that the optical power at the input PD(PD THORLABS RXM42AF) is 5 dBm, which is in the linear area of the PD. The variable gain amplifier (VGA) on the RF board is used to balance the amplitude of the two sub-arrays. The $S_{21}$ phase response of each link is obtained experimentally by detecting the signals fed back from the different output points (e.g., Figure 6, green monitoring point).

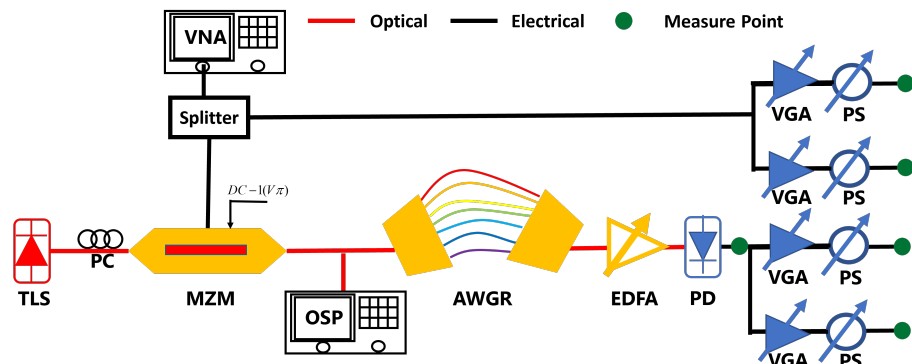

**Figure 5.** Schematic diagram of the measurement setup. TLS: tunable laser source; MZM: Mach–Zehnder modulator; PC: polarization controller; OSP: optical spectroscope; VNA: vector network analyzer; EDFA: erbium-doped fiber amplifier; AWGR: arrayed waveguide gratings router; PD: photodiode; VGA: variable gain amplifier; PS: phase shifter.

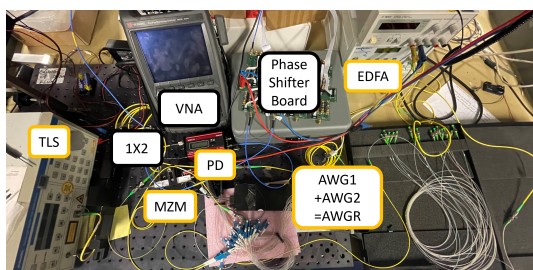

**Figure 6.** Schematic diagram of the measurement setup.

Wavelength switching-over will switch the optical carrier into a different channel of AWGR to obtain an adjustable true time delay. The phase response obtained at the measuring point PD allows for analyzing the delay generated by the different wavelengths of optical carriers passing through the AWGR. As illustrated in Figure 7 by fitting the data, different delays of AWGR can be obtained by phase vs. frequency slope: 0.058434 µs, 0.71755 µs, 0.91651 µs, 2.736 µs, 1.5178 µs, 3.3476 µs. Due to the instability of the PD, there are 5 *p*s errors in the delay time.

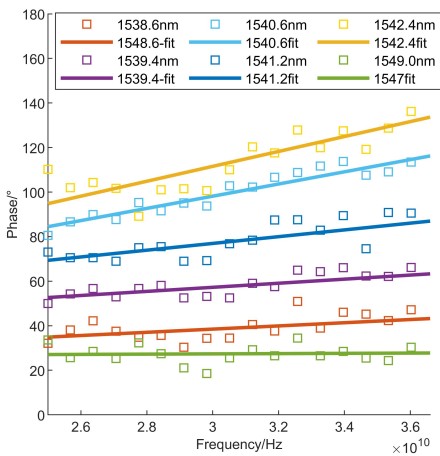

**Figure 7.** Phase response of different wavelength.

### 4.1. Overcome the Aperture Effect

In order to conduct far-field simulation experiments to verify the structure, 27 GHz is used as the center frequency of the RF signal and the bandwidth of the RF signal is 2 GHz.

The antenna array is ideally designed for the center frequency. The ideal antenna consists of four linear antenna arrays with equal spacing of $\lambda/2$, $\lambda = 27$ GHz/c. In Figure 8, the digital phase shifter is given a different amount of phase shift by the beam-forming principle. Simulation results show, without OTTD, beams shift to $-15.84°$ at 26 GHz and $-20.16°$ at 28 GHz. A beam pointing error of $4.32°$ occurs in the beam over a bandwidth of 2 GHz. The experimental results were similar, the beam points at $-14.76°$ and $-22°$ at 26 GHz and 28 GHz, respectively. This has a large pointing error. As shown in Figure 9, with the introduction of half of the OTTD in the link, the simulation results show that the beam is no longer shifted by large angles and the total beam error is less than $1°$. The phase response of each path was remeasured using the experimental setup. The $S_{21}$ phase response of each path was remeasured using the experimental equipment. After re-calculating each phase for beam synthesis, the beam is no longer shifted at 26 GHz, 27 GHz, and 28 GHz. In the AWGR-assisted phased array system proposed in this paper, signals with a transmission bandwidth of 2 GHz will no longer have a beam squint.

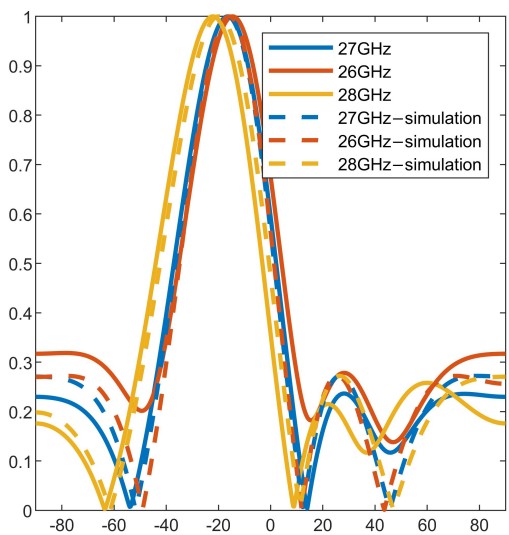

**Figure 8.** Beam squint with non-true time delay.

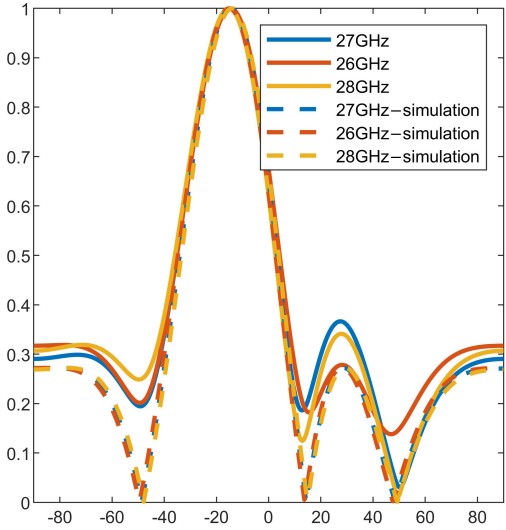

**Figure 9.** Non-beam squint with optical true time delay.

### 4.2. Wavelength Control Beam-Hopping and Fine Steering

In order to verify beam-hopping in multi-hop bent-pipe LEO satellites, the TLS is employed to change the wavelength across a broad range and superimpose various delay lines on the two subarrays in order to carry out five big angle beam scans, corresponding to −40°, −20°, 0°, 20°, and 40° in Figure 10. By comparing the simulation results with the experimental results, there is an experimental beam pointing error of <0.1°. The pointing error is due to the fact that the high-precision RF phase shifter used still has an accuracy error with a minimum phase shift step of approximately 5.6° ($360/2^6$).

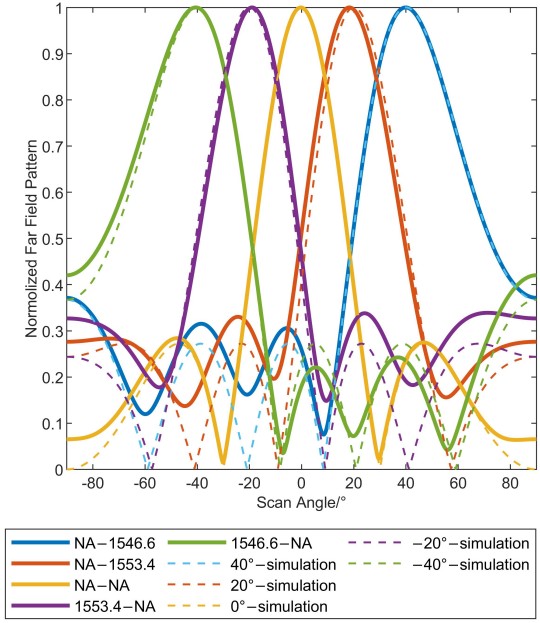

**Figure 10.** Far-field pattern of wide-angle steering.

By verifying 10, more precision beam pointing at a large angle is achieved. First, the measured results are calculated to synthesize the antenna far-field pattern as shown in Figure 11 when both subarrays are not supplemented with additional delay, the minimum interval of beam steering is 1.8° by digital phase shifter for phase compensation only, which achieves precise beam-pointing control. As shown in Figure 12b, combining the aforementioned two adjustment techniques allows for the modification of precise angles across a wide scanning range. Using a digital phase shifter to precision tune the beam, the beam in Figure 12a is scanned to point at −37.08°, −38.16°, −38.88°, −39.96°, and −40.68°. The beam is first aimed at about −40° by setting the wavelength to 1546.6nm. In a similar manner illustrated in Figure 12c, alter the wavelength so that the beam points at approximately 40°, then adjust the digital phase shifter so that the beam also points at 37.08°, 37.8°, 38.16°, 39.24°, and 39.96°. Thus, such a two-stage structure enables fast large-angle switching using a wavelength-controlled OTTD while also enabling fine-angle steering and target tracking using a digital phase shifter.

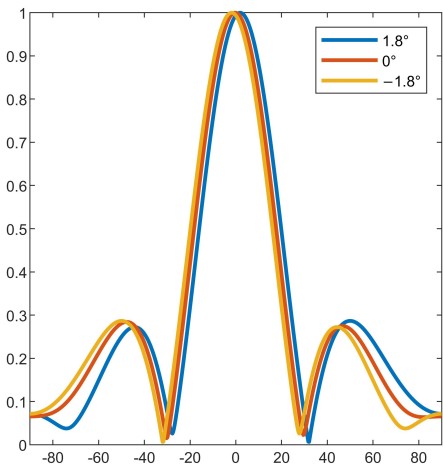

**Figure 11.** Far-field pattern of small angle steering.

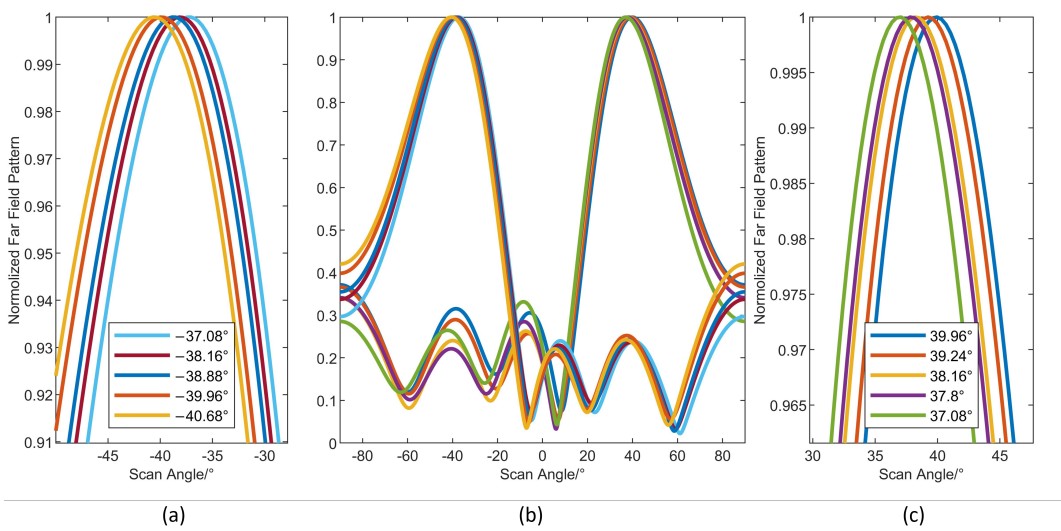

(a)                    (b)                    (c)

**Figure 12.** Far-field pattern of wide and fine beam steering (**a**) Scan details around −40°; (**b**) accurate angular scanning over a wide range of ±40° (**c**) scan details around 40°; (**c**) Scan details around 40°.

## 5. Discussion

As shown in Table 2, this study examines current work on OTTD implementation utilizing optical devices in order to evaluate the uniqueness and effect of such research in the larger area. Due to the challenge of producing ultra-high precision optical time delays using optical chips in current techniques, it is challenging to steer the beam for phased arrays utilizing complete OTTD with high accuracy. In 2018, Xiang described the optical chip with the highest precision delay accuracy. With a minimum delay regulation compensation of 0–3.4 ps, it has a construction that is an ultra-low-loss continuously tunable optical real-time delay line device based on a Si3N4 ring resonator (MRR) in a side-coupled integrated spaced sequence resonator (SCISOR) structure [44]. A beam-steering precision of 0 to 3.8991° is consistent with such a delay accuracy. The MRR, however, has outstanding spectrum properties and is particularly temperature-sensitive; as a result, when the temperature changes, the device's response alters significantly.

**Table 2.** Comparison between different OTTD and array.

| Example | Beam-Steering Precision° | Scan Angle° | Frequence/GHz |
|---|---|---|---|
| MRR [44] | 0 3.8991 | - | 10 |
| GDL [45] | 5.393 | - | 10 |
| Optical Switch [46] | 2.2924 | $\pm 75$ | 8–18 |
| Comb [47] | 22.0243 | $\pm 60$ | 8–20 |
| This paper | 1.8 | $\pm 40$ | 27 |

The grating delay line (GDL) is another OTTD option that is simpler to integrate. Its minimum time delay depends on the special design of the grating. The present structure has a minimum beam-steering precision of 5.393° and uses waveguide gratings as the OTTD for beam synthesis [45]. Nevertheless, because of limitations such as a lack of process precision, the lowest beam gaze accuracy that can be achieved with optical switches and Comb for OTTD is 2.2924° and 22.0243°, respectively [46,47].

In order to accomplish high-precision beam steering while accounting for subarray delays, this work ingeniously integrates a high-precision millimeter beamforming device with an OTTD that is AWGR-based.

## 6. Conclusions

An innovative low-orbit satellite network design is put forth in this research. With this design, beam hopping, a crucial regenerative network method, is combined with the benefits of non-regenerative networks. It is the first architecture to support a multi-hop bent-pipe LEO satellite network employing beam hopping in the DVB-S2X protocol. To increase the adaptability of satellite constellation networks, it incorporates beam-hopping into the inter-satellite routing algorithm of LEO satellites. To enable this design's deployment, it also uses AWGR devices. The test results of the vector network analyzer verify 1. a decrease of the phased-array antennas' beam pointing offset during broad-band wide-angle scanning; 2. wavelength control of the antenna pointing direction; and 3. high-precision beam steering over a wide range of scanning angles. This research has great potential application possibilities, it enhances the flexibility of bent-pipe LEO satellite network architecture, reduces the weight of a single satellite, and ensures user data security.

**Author Contributions:** Conceptualization, S.X., M.Z., P.L. and G.L.; methodology, S.X., M.Z.; experiment, S.X. and M.Z.; validation, S.X., M.Z. and P.L.; investigation, Z.C., F.S., J.J. and H.W.; resources, P.L. and G.L.; writing—original draft preparation, S.X., P.L., M.Z., F.S. and Z.C.; writing—review and editing, M.Z., P.L., C.Z., F.S., J.J., and H.W.; supervision, G.L.; project administration, G.L.; funding acquisition, G.L. and M.Z. All authors have read and agreed to the published version of the manuscript.

**Funding:** This research was funded by the National special support plan for high-level talents (no. WRJH19DH01), and the China Postdoctoral Science Foundation (2020M682863).

**Institutional Review Board Statement:** Not applicable.

**Informed Consent Statement:** Not applicable.

**Data Availability Statement:** The data presented in this study are available on request from the corresponding author.

**Conflicts of Interest:** The authors declare no conflict of interest.

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
