# Peer review of "Large Scale Optical Assisted Mm-Wave Beam-Hopping System for Multi-Hop Bent-Pipe LEO Satellite Networks"

_applsci, doi:10.3390/app13063480_

Round 1
Reviewer 1 Report
- Fig. 3, Fig. 4 It might be better to increase the font size.
- It would be interesting to add the array information (array shape, subarray number (M), elements in a subarray (m), array distance (d), total element number, etc).
- Table 1 It would be nice to add the explanation to the angle and unit to the parameters.
- It would be better to add unit to the beamforming angle (line 233, 234, 245, etc).
- Fig. 8 Is the normalized pattern power or voltage level? At 26, 27 GHz, the side lobe looks very high with or without OTTD. It would be better to explain the reason. It would be recommended to mention the center frequency and beam pointing error within the bandwidth.
- Fig. 8 What type of element pattern was assumed for beam calculation?
- Fig. 11 There is a caption error ((a) scan details around 40 -> (c) scan details around 40). It would be recommended to explain Fig. 11(b) in the figure caption.
- line 257 It would be interesting to add the simulated or measured switching time.
Author Response
Dear Referee:
Thank you for taking the time to review my work. I really appreciate your feedback and am grateful for your valuable insights.
I have taken your comments into consideration and have made the following changes to my work:
Advice1. Fig. 3, Fig. 4 It might be better to increase the font size.
Reply: Thanks to your suggestion, I have added Font Size in Fig. 3, Fig. 4. Please refer to the yellow marker in the revised version
Advice2: It would be interesting to add the array information (array shape, subarray number (M), elements in a subarray (m), array distance (d), total element number, etc).
Reply: I added a description of the antenna array for simulation and experiment. The antenna array is ideally designed for the centre frequency 27GHz. The ideal antenna consists of four antenna arrays with a equally spacing of λ/2, λ = 27GHz/c. Please refer to the pink marker in the revised version.
Advice3: Table 1 It would be nice to add the explanation to the angle and unit to the parameters.
Reply: Thanks for your suggestion, I have added the angle unit parameter in Table1. Please refer to the green marker in the revised version.
Advice4: It would be better to add unit to the beamforming angle (line 233, 234, 245, etc).
Reply: I have added the angle unit parameter. Please refer to the green marker in the revised version.
Advice5: Fig. 8 Is the normalized pattern power or voltage level? At 26, 27 GHz, the side lobe looks very high with or without OTTD. It would be better to explain the reason. It would be recommended to mention the center frequency and beam pointing error within the bandwidth.
Reply: Thanks for your suggestion, I have corrected the Fig8. This is due to the unbalanced amplitude of each output port. After I re-balanced the channels of the board, this phenomenon disappeared, the side lobe appeared too high, which has now been corrected. The normalized pattern is power . And I added some explanation of the center frequency, and beam pointing error within the bandwidth. Please refer to the blue marker in the revised version.
Advice6:Fig. 8 What type of element pattern was assumed for beam calculation?
Reply: I added a description of the antenna array for simulation and experiment. The antenna array is ideally designed for the centre frequency 27GHz. The ideal antenna consists of four antenna arrays with a equally spacing of λ/2, λ = 27GHz/c. Please refer to the pink marker in the revised version.
Advice7: Fig. 11 There is a caption error ((a) scan details around 40 -> (c) scan details around 40). It would be recommended to explain Fig. 11(b) in the figure caption.
Reply: I have corrected caption error in Fig. 11
Advice8: line 257 It would be interesting to add the simulated or measured switching time.
Reply: have added simulation experiments. Please refer to Fig8, Fig9 and Fig.10. Due to the limitation of the experimental equipment, the testing of the switching time is not feasible. However, since AWGR is a passive device, the wavelength switching time is negligible. It only depends on the laser wavelength switching time. For the millimeter wave beamforming chip, according to the parameters in the datasheet, it is possible to achieve a switching speed of 10 ns with a pre-set 16-group phase combination.
I hope that these changes have addressed your concerns.
Once again, thank you for your time and expertise.
Sincerely,
Shiyi

Reviewer 2 Report
In this communication manuscript, an innovative low-orbit satellite network design is presented. With this design, beam hopping, a crucial regenerative network method, are combined with the benefits of non-regenerative networks. The reviewer recommends minor revision considering the following comments:
1- The Abstract is obscure, it does not reflect the contribution of the manuscript;
2- Some Tables and Figures are presented before being mentioned in the text;
3- English Proofreading is required by native speakers.
Author Response
Dear Referee:
Thank you for taking the time to review my work. I really appreciate your feedback and am grateful for your valuable insights.
I have taken your comments into consideration and have made the following changes to my work:
Advice1: The Abstract is obscure, it does not reflect the contribution of the manuscript;
Reply: I have revised Abstract. Please refer Abstract.
Advice2: Some Tables and Figures are presented before being mentioned in the text;
Reply: I have add the quote of Fig.1. Please refer Blue mark.
Advice3: English Proofreading is required by native speakers.
Reply: I have carefully rephrase this paper. Please refer Pink mark.
I hope that these changes have addressed your concerns.
Once again, thank you for your time and expertise.
Sincerely,
Shiyi

Reviewer 3 Report
Although the research presented in this paper is innovative, there are opportunities to enhance the technical rigor of the analysis. Specifically, it is suggested that the authors conduct a comparison between the simulation and measurement results for the far field pattern to validate the accuracy of the theoretical model. This would enable the authors to identify any discrepancies between the simulated and measured data, and provide insights into potential sources of error. Additionally, we recommend that the authors include a comparative analysis of their results against other studies in the literature, utilizing a detailed comparison table. This would enable readers to more comprehensively evaluate the novelty and impact of the research within the broader context of the field. By addressing these suggestions, the paper would be improved and would make a more significant contribution to the field.
Author Response
Dear Referee:
Thank you for taking the time to review my work. I really appreciate your feedback and am grateful for your valuable insights.
I have taken your comments into consideration and have made the following changes to my work:
Advice1: Specifically, it is suggested that the authors conduct a comparison between the simulation and measurement results for the far field pattern to validate the accuracy of the theoretical model. This would enable the authors to identify any discrepancies between the simulated and measured data, and provide insights into potential sources of error.
Reply: Thanks for your suggestion, I have added simulation experiments. Please refer to Fig8, Fig9 and Fig.10, and yellow mark in the revised paper.
Advice2: Additionally, we recommend that the authors include a comparative analysis of their results against other studies in the literature, utilizing a detailed comparison table.
Reply: Thanks for your suggestion, I have added comparative analysis in the Discussion section. Please refer blue mark in the revised paper.
I hope that these changes have addressed your concerns.
Once again, thank you for your time and expertise.
Sincerely,
Shiyi
